# Engagement in Health Risk Behaviours before and during the COVID-19 Pandemic in German University Students: Results of a Cross-Sectional Study

**DOI:** 10.3390/ijerph18041410

**Published:** 2021-02-03

**Authors:** Heide Busse, Christoph Buck, Christiane Stock, Hajo Zeeb, Claudia R. Pischke, Paula Mayara Matos Fialho, Claus Wendt, Stefanie Maria Helmer

**Affiliations:** 1Leibniz Institute for Prevention Research and Epidemiology-BIPS, 28359 Bremen, Germany; buck@leibniz-bips.de (C.B.); zeeb@leibniz-bips.de (H.Z.); 2Institute of Health and Nursing Science, Charité—Universitätsmedizin Berlin, Corporate Member of Freie Universität Berlin, Humboldt-Universität zu Berlin, and Berlin Institute of Health, Augustenburger Platz 1, 13353 Berlin, Germany; christiane.stock@charite.de (C.S.); stefanie.helmer@charite.de (S.M.H.); 3Health Sciences Bremen, University of Bremen, 28359 Bremen, Germany; 4Institute of Medical Sociology, Centre for Health and Society, Medical Faculty, Heinrich Heine University Duesseldorf, 40225 Duesseldorf, Germany; ClaudiaRuth.Pischke@med.uni-duesseldorf.de (C.R.P.); PaulaMayara.MatosFialho@med.uni-duesseldorf.de (P.M.M.F.); 5Department Sociology of Health and Healthcare Systems, University Siegen, 57068 Siegen, Germany; wendt@soziologie.uni-siegen.de

**Keywords:** university students, COVID-19, health-risk behavior, alcohol use, smoking, cannabis, physical activity, pandemic

## Abstract

Tobacco and cannabis use, alcohol consumption and inactivity are health risk behaviors (HRB) of crucial importance for health and wellbeing. The impact of the COVID-19 pandemic on university students’ engagement in HRB has yet received limited attention. We investigated whether HRB changed during the COVID-19 pandemic, assessed factors associated with change and profiles of HRB changes in university students. A web-based survey was conducted in May 2020, including 5021 students of four German universities (69% female, the mean age of 24.4 years (SD = 5.1)). Sixty-one percent of students reported consuming alcohol, 45.8% binge drinking, 44% inactivity, 19.4% smoking and 10.8% cannabis use. While smoking and cannabis use remained unchanged during the COVID-19 pandemic, 24.4% reported a decrease in binge drinking while 5.4% reported an increase. Changes to physical activity were most frequently reported, with 30.6% reporting an increase and 19.3% reporting a decrease in vigorous physical activity. Being female, younger age, being bored, not having a trusted person and depressive symptoms were factors associated with a change in HRB. Five substance use behavior profiles were identified, which also remained fairly unchanged. Efforts to promote student health and wellbeing continue to be required, also in times of the COVID-19 pandemic.

## 1. Introduction

### 1.1. Engagement in HRB among the University Student Population

Lifestyle behaviors, such as tobacco use, excessive alcohol consumption, cannabis use and physical inactivity are health-risk behaviors (HRB) of crucial importance for health and wellbeing. As key factors in the prevention of non-communicable diseases and primary causes of premature morbidity and mortality [1], engagement in HRB carries high costs to individuals and society. While engagement in HRB is typically initiated in the adolescent years, engagement is continued into “emerging adulthood” [2], where for many young people, the transition from home and secondary school into higher education or the labor market is made. The transition to university, in particular, often goes alongside a change of social networks, environments, and increased freedom, aspects that have been brought into connection with engagement in HRB [3]. Although most evidence on high rates of engagement in HRB in university students is available from North America [4], HRB among the university student population has also been consistently reported in Europe and elsewhere [5,6].

Alcohol intake is the leading cause of concern in university students [7,8], and binge drinking has lately received particular attention due to its acute and long-term negative consequences [9]. Binge drinking peaks in late adolescence or early adulthood and is, therefore, a relevant HRB in this group [9]. In addition, emerging adults show an increased risk of tobacco addiction, and this tendency was found to be independent of whether individuals smoked at secondary school or not [10]. With regard to cannabis use, different trajectories were reported previously. More than 70% of students remained non-users of cannabis throughout their studies or kept cannabis use at a low-level (10%), but the third-largest group of students increased their use significantly during their studies [11]. Lastly, a large percentage of the student population does not engage in enough physical activity (PA) [12]. In a review of 19 primary studies, 60% of university students were found to be physically inactive [13]. A study assessing whether or not Portuguese university students achieved PA recommendations using accelerometers showed more inactivity reported during the weekend and female students being more sedentary than male students [14].

While comparably little research exists on the health and wellbeing of the German university student population, previous research indicates that risk behaviors, such as harmful substance use, are prevalent among students at German universities. In a survey among students of 16 universities in the German federal state of North Rhine-Westphalia, over 90% of students reported consuming alcohol, with 80% reporting heavy drinking [15]. Over 60% of the surveyed students had never smoked, 15% were classified as former, and 23% as current smokers. Moreover, 41% of students had used cannabis at least once in their lifetime, 9% reported consuming cannabis within the last 30 days [16]. A study focusing on first-year university students in Germany reported that 31% were smokers, 62% reported binge drinking, and 60% did not exercise sufficiently [17]. Similarly, a reduction of PA in the transition from high school to university was reported in a German sample of university students [3].

### 1.2. Characteristics Associated with Engagement in HRB

A multitude of factors have been shown to be related to engagement in HRB, spanning across sociodemographic (e.g., age, sex), individual (e.g., motivation, attitudes), social (e.g., social networks, peer influences, social norms), environmental (e.g., resources, context, physical environment) and macrolevel factors (e.g., governmental regulations). Examples of sociodemographic influences are that male students tend to consume alcohol more regularly and in higher quantities compared to female students [18]. While no gender differences in terms of smoking are typically reported, men also have higher rates of cannabis use [19]. Additionally, mental health has been linked to substance abuse behaviors [20]. In a study among university students of seven European countries, poor mental health was reported as a risk factor for problematic alcohol use [21]. Medical students have previously been found less likely to engage in HRB compared to students studying other subjects [17]. This has been argued to be due to their self-selection into a more health-conscious study subject and profession [17].

HRBs tend to cluster, meaning that engagement in one risk behavior is likely to co-occur with engagement in another risk behavior [22,23]. In a systematic review of 37 studies, alcohol misuse and smoking were found to be the most commonly identified risk behavior cluster [23], highlighting how particularly substance use behaviors are closely interlinked.

### 1.3. The COVID-19 Pandemic

The SARS-CoV-2 outbreak and the corresponding COVID-19 pandemic have had and continue to have a significant impact on the health and wellbeing of the general population globally and in Germany. Various large-scale preventive measures were implemented by governments across the world, which led to significant changes in daily life. University students are in a unique position during the COVID-19 pandemic, as they are confronted with substantial changes and restrictions, both at the governmental and university-level. A recent investigation by Aristovnik and colleagues [24], analyzing data from an international sample of over 30,000 higher education students, highlights the stark changes that students experience in their social and academic life during the COVID-19 pandemic. Students were found to experience boredom, anxiety, and frustrations because of the COVID-19 pandemic and expressed concerns about their future professional careers and studies [24].

Research on earlier epidemics and financial crises highlights that such external events may impact one’s engagement in HRB. A review examining the impact of the 2008 Economic Crisis on substance use found a bidirectional effect, with several studies reporting a decreased overall substance use in the general population, but also an increasing harmful use within specific subgroups (e.g., unemployed) [25]. One study conducted on the SARS epidemic in Hong Kong reported that both medical and non-medical students reported higher levels of anxiety and stress [26]. In a review of the existing evidence, quarantine was shown to be associated with negative effects, such as confusion and anger, and stressors, such as frustrations, boredom, financial loss, and stigma, with the potential for long-lasting effects [27].

Some initial studies on the impact of the COVID-19 pandemic on university student behavior and feelings have been published. In a study of alcohol consumption in times of COVID-19 among adults the general population in 21 European countries, alcohol consumption was found to have decreased on average [28]. Interestingly, alcohol consumption in Germany was found to have declined less sharply due to an increase in consumption among women as well as those who reported negative effects of the pandemic (e.g., on jobs or finances) and those with risky consumption patterns [28]. A study among more than 3000 university students in Turkey reported that more than one in three students reported worrying about COVID-19 [29]. Similarly, a study among medical students at a Chinese university reported a positive association between anxiety symptoms and economic hardship and effects on daily life [30]. Those who had to self-isolate due to COVID-19 were found to consume more cannabis leading authors to conclude that self-isolation is a risk factor for cannabis use [30].

To our knowledge, there is no study that has yet investigated the impact of the COVID-19 pandemic, and more specifically, the lockdown period, on engagement in HRB among university students in Germany.

### 1.4. Study Questions

We conducted an online survey during the first wave of restrictions of the COVID-19 pandemic in a sample of students from four German universities to investigate the following three research questions:Is there any change of engagement in HRB in German university students during the COVID-19 pandemic?What characteristics are associated with a change of engagement in HRB during the COVID-19 pandemic among German university students?Which profiles of engagement in substance use behaviors and changes among profiles can be identified?

## 2. Materials and Methods

This study is part of the COVID-19 International Student Well-Being Study (C19 ISWS) [31]. C19 ISWS is the result of a study design, study protocol and questionnaire developed by a team of the University of Antwerp, Belgium (Prof. Sarah Van de Velde, Dr. Veerle Buffel and Prof. Edwin Wouters). C19 ISWS is an international project running at multiple universities during the COVID-19 pandemic.

### 2.1. Participants

An online survey was conducted among university students at four participating German universities (the University of Bremen, University of Siegen, Charité—Universitätsmedizin Berlin, Heinrich Heine University Duesseldorf). Participants who were currently enrolled as students (Bachelor, Master, state exam, PhD or other) and who were aged 17 years and above were eligible to participate. The recruitment goal outlined in the study protocol of the overarching C19 ISWS was to sample at least 10% of the student population at each participating university.

### 2.2. Online-Survey

The survey questions were originally developed by the consortium leaders based on work by Brooks and colleagues [27]. Questions were refined in the international consortium, and each country had the possibility to add questions to the survey. In line with the translation protocol, the survey was translated independently by two authors of the German team (S.M.H., H.B.). The translation was discussed in detail, and any disagreements were resolved by reaching a consensus.

Participants’ sociodemographic characteristics were captured at the beginning of the survey, followed by participants’ study-related information, their living situation before and during the COVID-19 outbreak, engagement in HRB, COVID-19 diagnosis, COVID-19 symptoms, perceived worries, mental wellbeing, participants’ knowledge about COVID-19 and critical health literacy towards COVID-19. The core questionnaire used is publicly available [32].

### 2.3. Data Collection and Context

Data were collected in spring within a period of two weeks at each university, spanning the period from 12 to 29 May 2020 in the German survey. The German government announced severe restrictions (termed “lockdown” in this paper) on 22 March for the whole of Germany, which included the closure of universities (including restaurants, libraries, and cafeterias on campus), non-essential shops, restaurants and other services, and the introduction of social distancing requirements. From mid-April onwards, some restrictions were loosened, and some buildings, such as museums, monuments, botanical gardens, and zoos, were opened, as well as religious services allowed under strict social distancing conditions. The participating higher education institutions canceled most in-person classes and offered online classes. Access to university buildings was restricted, and all libraries, as well as student restaurants and cafeterias, remained closed. Furthermore, student counseling had to change their services into online services.

University-wide email distribution lists were used to invite students to participate in the study in all but one university. Additionally, some universities posted notifications on the main university homepage, e-learning platforms and similar websites to recruit participants. One university solely recruited students via the faculty email list and social media. If interested in participating, students were asked to follow a link to the survey website, where the option was given to complete the survey in German or English. Detailed study information was provided, and students were asked for their consent to take part in the survey. Approximately seven days after the start of data collection, a reminder email was sent out. A list of country-specific resources for students was provided at the beginning and end of the survey should they wish to speak to someone about their feelings.

### 2.4. Data Management

Data collection was coordinated by the consortium leaders with the use of a Qualtrics^®^ (Qualtrics, Provo, UT, USA) survey. The survey was administered through a secure, web-based platform hosted by the University of Antwerp. All participating universities signed contracts with the consortium leader to ensure data protection principles were followed.

### 2.5. Ethical Approval

Ethical approval for the study was obtained from the ethics committees of each participating university in Germany. All participants provided informed consent to take part in the survey.

### 2.6. Measures

#### 2.6.1. HRB

The outcomes of interest for HRB were: (1) number of cigarettes smoked, (2) number of drinks consumed, (3) binge drinking, (4) cannabis use and, (5) engagement in vigorous PA and, respectively (6) moderate PA. Selected risk behaviors were assessed retrospectively and at the time of data collection to examine situation-specific changes prior to and during the COVID-19 pandemic. For retrospective assessment, the following statement was used: with “before the COVID-19 outbreak”, we refer to the average situation during the month prior to the moment that the first COVID-19 measures (e.g., social distancing measures) were implemented. For assessment of behaviors during the COVID-19 pandemic, the following statement was used: with “during the last week”, we refer to the week prior to filling out this survey.

Number of cigarettes per day—Students were asked to indicate the total number of cigarettes smoked, on average, per day prior to and during COVID-19. Smoking was categorized as “none”, “1–9 cigarettes per day”, and “10 or more cigarettes per day”.

Number of drinks per week—Students were asked to indicate the total number of drinks per week prior to and during the COVID-19 outbreak. In a statement placed prior to the alcohol-related questions, students were informed that a drink was defined as a glass of wine, a shot, a glass of beer between 25 to 33 centiliters (cl). The number of drinks was categorized as “none”, “1–2 drinks”, and “3–5 drinks” or “5 or more drinks”.

Binge drinking—Drinking six or more glasses of alcohol on a single occasion was defined as binge drinking, in line with the wording used in the “Alcohol Use Disorders Identification Test” (AUDIT) [33]. The frequency of heavy drinking occasions was assessed using the question “How often do you have six or more drinks on a single occasion?”. Response options ranged on a 5-point scale from “(almost) never” and “less than once a week” to “(almost) daily”. The frequency of binge drinking was categorized as “(almost) never”, “low” (less than once a week/once a week), and “high” (more than once a week/(almost) daily).

Cannabis use—Students were asked, “on average, how often did you use cannabis (marijuana, weed, hash, …)?”. Response options ranged on a 5-item scale from “(almost) never” and “less than once a week” to “(almost) daily”. The frequency of cannabis consumption was categorized as “(almost) never”, “low”, and “high”, similar to the frequency of binge drinking.

Vigorous PA (VPA)—Students were asked how frequently they performed vigorous physical activities, such as lifting heavy things, running, aerobics, or fast cycling for at least 30 min. Moderate PA (MPA): Students were asked how often they performed moderate physical activities, such as easy cycling or walking for at least 30 min. Response options for VPA and MPA ranged from “(almost) never” and “less than once a week” to “(almost) daily”. For both PA questions, the same response categories were employed for binge drinking and cannabis use, and data were categorized into “(almost) never”, “(less than) once per week,” and “more than once a week/(almost) daily”.

To indicate whether students changed their HRB during COVID-19 compared to prior to the COVID-19 pandemic, we compared the categorized HRB estimates of the respective variables. The change in engagement in HRB was collapsed into three categories “increase”, “no change”, and “decrease”.

#### 2.6.2. Covariates

Sociodemographic data—Participant’s age, gender (female, male, diverse), place of birth (born in Germany or not), relationship status (single, in a relationship, “it is complicated”) were assessed. Where age was used in a categorized manner, it was collapsed into the following categories: “17–18”, “19–20”, “21–22”, “23–24”, and “25 and older”.

Study-related information—Students were asked to indicate whether they were first-year students (yes/no) and which study subject they were enrolled in (health sciences or medicine versus other study fields or more than one field).

Depressive symptoms—Depressive symptoms were measured using the 8-item Center for Epidemiologic Studies—Depression Scale (CES-D 8), which was previously validated as a measure among the general population to identify populations at risk of developing depression [34]. In the CES-D 8, respondents are asked how often in the previous week they felt “depressed”, “lonely”, “sad”, “happy”, “enjoyed life”, “felt everything they did was an effort”, had “restless sleep” and “could not get going”. Response options on the 4-point Likert scale ranged were (0) “none or almost none of the time”, (1) “sometimes”, (2) “most of the time” to (3) “all or almost all of the time”. The two items, “happy” and “enjoyed life,” were reverse coded, and scores were summed.

Trusted other—To assess access to a trusted other person, students were asked, “Do you have anyone with whom you can discuss any intimate and personal matters?” (yes/no).

Boredom: Students were asked to indicate how much of the time during the past week they were bored. The response options ranged from “(almost) none of the time”, “some of the time”, “most of the time” and “(almost) all of the time” were categorized into three categories “none”, “some” and “most/all”.

### 2.7. Analysis

Descriptive analysis was performed using tabulations for HRB. Furthermore, percentages of respondents who reported an increase/no change/decrease in HRB before and during the COVID-19 pandemic were calculated. As the dependent variables consisted of three groups (increased/no change/decreased HRB), multinomial logistic regression analyses were conducted to identify the associations between the independent and the respective dependent variables. Age, gender, place of birth, relationship status, first-year student, study subject, depressive symptoms, trusted other, and boredom were included as independent variables in the multinomial regression models. For these analyses, age was used as a continuous variable and all others as categorical variables.

In order to identify profiles of HRB in participants and changes within profiles, a latent transition analysis (LTA) among substance use behavior (smoking, binge drinking, and cannabis use) was conducted. Based on latent class analysis, the LTA allows the calculation of transition probabilities for latent classes [35,36]. Equal numbers of latent classes were calculated for risk behaviors, i.e., categories of the number of cigarettes, frequency of binge drinking, and frequency of cannabis consumption, for both time points (i.e., retrospective pre-COVID-19 pandemic and during the COVID-19 pandemic). LTA models with three to six latent classes were estimated, and the best model fit was evaluated based on the Bayesian information criterion (BIC). For the selected model, latent classes were labeled and characterized based on the highest item response probabilities (ρ-estimates) and transition probabilities.

Descriptive statistics and multinomial regressions were conducted in SPSS Version 26.0 (International Business Machines Corporation (IBM), Armonk, NY, USA). PROC LTA Version 1.3.2 in SAS Version 9.4 (SAS Institute, Cary, NC, USA) was used to conduct the latent transition analysis.

## 3. Results

### 3.1. Description of Participants

A total of 5021 university students completed the survey, of which 69% were female and 29% male with a mean age of 24.4 years (SD = 5.1). The mean score for depressive symptoms based on the CESD-8 was 9.25 (SD = 0.07). A description of the study sample is provided in Table 1. The intention to sample at least 10% of students was achieved in three of the four participating German universities. The approximated response rates for three of the universities were 10–11%. In the university that solely recruited via email in one of five faculties (Medical Faculty) and which used social media for recruitment of the overall student population at the university, a response rate of approximately 2% was reached at the university level and a response rate of approximately 17% at the faculty level.

### 3.2. Engagement in HRB and Changes Prior and during COVID-19

The two most commonly reported risk behaviors were alcohol consumption and physical inactivity. Around 19% of students reported smoking, with 13% smoking cigarettes more than once per week before and during the COVID-19 pandemic. Prior to the COVID-19 pandemic, 85% percent did not smoke any cigarettes, 10% 1–9 cigarettes, 4% 10–19 cigarettes and 1% more than 20 cigarettes per day. These proportions did not change substantially during the pandemic.

While 2% of students reported binge drinking more than once per week prior to the COVID-19 pandemic, 4% of students reported binge drinking during the COVID-19 pandemic. With regard to the number of alcoholic drinks consumed, 39% reported zero alcoholic drinks per week on average prior to COVID-19, 34% 1 to 2 drinks, 18% 3–5 drinks and 10% 6 and more drinks. During the COVID-19 pandemic, 46% reported drinking zero drinks, on average, 26% 1–2 drinks, 17% 3–5 drinks and 12% 6 and more drinks, indicating little change.

While 51% of students reported performing VPA more than once a week before the COVID-19 pandemic, this percentage was 45% during the pandemic (Table 2).

Whether changes were reported during the COVID-19 pandemic in comparison to before COVID-19 differed by the type of HRB investigated (Table 3). While smoking and cannabis use were found to be unchanged in the majority of the respondents (92% for smoking, 93% for cannabis use), 38% reported changes in their consumed number of alcoholic drinks per day, and a further 30% reported changes in their binge drinking. While 18% reported an increase in the number of alcoholic drinks consumed, 19% reported a decrease. With regard to binge drinking, 24% reported a lower level of binge drinking, while 5% reported an increase in binge drinking. The highest change was reported with regard to engagement in PA. While up to 55% showed identical PA behavior before and during the pandemic, 31% of students reported a decrease in VPA and 22% in MPA. However, 19% of students reported an increase in VPA and 23% an increase in MPA.

### 3.3. Characteristics Associated with Change in Engagement of HRB

Reporting depressive symptoms was found to be associated with increased HRB (number of cigarettes and drinks, binge drinking as well as cannabis use) but was also associated with a decrease in tobacco use during the COVID-19 pandemic. Being in a complicated relationship was associated with a higher chance of an increase in the number of cigarettes smoked (OR: 2.03, 95% CI: 1.15–3.57), the number of drinks consumed (OR: 1.56, 95% CI: 1.07–2.28), and binge drinking (OR: 1.77, 95% CI: 1.03–3.04), but also with a higher chance for a decrease in the number of cigarettes smoked (OR: 2.35, 95% CI: 1.29–4.29). Additionally, students who reported being bored most or all of the time were more likely to show a decrease in the number of cigarettes (OR: 1.74, 95% CI: 1.15–2.64), and those who reported being bored some of the time showed an increase in the frequency of binge drinking (OR: 1.38, 95% CI: 1.02–1.87). Female and single students reported an increase in the number of drinks consumed. No differences were observed in changes in HRB among first-year students compared to non-first-year students as well as among students from a health-related subject compared to a non-health-related subject. Similarly, no differences were observed among individuals who had or did not report having a trusted person to speak to about personal matters (Table 4).

With regard to engagement in PA during the COVID-19 pandemic, depressive symptoms were also found to be associated with both an increase in VPA (OR = 1.02, 95% CI: 1.01–1.04) and MPA (OR = 1.03, 95% CI: 1.01–1.05) and a decrease in VPA (OR = 1.08, 95% CI: 1.06–1.09) and MPA (OR = 1.10, 95% CI: 1.09–1.12). Similarly, students who reported being bored most or all of the time were more likely to show an increase in MPA (OR: 1.23, 95% CI: 1.01–1.49), but also a decrease in VPA (OR: 1.42, 95% CI: 1.19–1.70) and MPA (OR: 1.35, 95% CI: 1.11–1.64). Compared to males, females were more likely to increase PA during the COVID-19 pandemic found for both VPA (OR = 1.44, 95% CI: 1.20–1.73) and MPA (OR = 1.32, 95% CI: 1.11–1.55). However, compared to males, females were also more likely to decrease MPA (OR = 0.77, 95% CI: 0.66–0.91). Younger students had a higher chance for increased levels of VPA (OR = 0.97, 95% CI: 0.96–0.99 for continuous age) and MPA (OR = 0.96, 95% CI: 0.94–0.97 for continuous age). Relationship status and being a first-year student were not found to be associated with changes in VPA or MPA. Individuals who did not report having a trusted person to speak to about personal matters were more likely to show a decrease in MPA (OR: 1.39, 95% CI: 1.09–1.76; Table 5).

### 3.4. Results of Latent Transition Analysis

LTA models showed the best-fit when considering five latent profiles (see Table 6). Following item-response probabilities, latent profiles were characterized as follows: (1) nonsmoker, infrequent binge drinker, (2) nonsmoker, infrequent binge drinker and cannabis consumer, (3) nonconsumer, (4) regular smoker and frequent cannabis consumer, and (5) regular smoker. The third profile showed the highest proportion prior to (47.7%) and during the COVID-19 pandemic (66.8%).

With respect to the transition matrix, diagonal elements reflect the proportion of individuals with the same behavior profile at both times. Nonconsumer was considered as the reference profile showing consistently no HRB prior to COVID-19, where 97.6% remained in this profile during the COVID-19 pandemic (see Table 6). Profiles 1 and 2 (including infrequent binge drinking) maintained HRB with only 43.4% (profile 1) and 62.9% (profile 2) during the COVID-19 pandemic, respectively. While for profile 1 (i.e., no cannabis consumption), the majority changed to the profile of nonconsumer (54.3%), profile 2, including cannabis consumption, 26.1% transitioned to nonconsumer (profile 3), and 11.0% transitioned to profile 4, including smoking and cannabis consumption. The majority of profiles, including regular smokers, remained unchanged either with (profile 4, 92.0%) or without cannabis consumption (profile 5, 96.4%).

## 4. Discussion

This study investigated students’ engagement in HRB before and during the COVID-19 pandemic at four German universities using the C19 ISWS dataset. We found that the majority of students reported engagement in two risk behaviors (alcohol consumption, physical inactivity), 19% reported tobacco smoking, and 11% reported cannabis consumption. Almost half of the students reported binge drinking prior to the COVID-19 pandemic. For the majority of students, no substantial changes in engagement in HRB during the COVID-19 pandemic compared to prior to the pandemic were noted. While smoking and cannabis use were found to be the most stable behaviors, alcohol consumption and physical activity were the behaviors that were reported to have changed the most among the German student population.

Tobacco use did not change substantially during the COVID-19 pandemic. Interestingly, a study among US-adults found an increase in individual’s motivation to quit tobacco use [37]. As we only captured student’s behavior, we are unable to make a statement about attempts to quit tobacco use amongst the general German population. Another behavior that remained stable in our university student sample was cannabis use. Again, our findings differed with regard to a study conducted among US adults during the COVID-19 restrictions, which found that 37% increased their cannabis use [38]. One in five students decreased, but the same proportion increased their consumed number of alcoholic drinks. A survey conducted among US adults arrived at different results, highlighting more change in this behavior: 49% of adults increased their alcohol consumption, whereas 12% were found to have decreased their consumption [38]. Almost one in four students reported binge drink less often during the COVID-19 pandemic, and only a few increased binge drinking. Binge drinking was thus the one behavior found to be less frequently reported during the COVID-19 pandemic, which is in line with previous studies [25,38] and can be explained by the social nature of this risk-taking behavior [26]. Approximately one-third of the surveyed students reported a decrease in PA, another third an increase in PA, highlighting how engagement in this lifestyle behavior varied considerably among students. These findings are in line with results from a survey among US-adults during the COVID-19 pandemic [38]. Restrictions due to COVID-19, such as pub, clubs and restaurant closures, as well as restrictions of the number of people allowed at social gatherings, may represent reasons for the changes identified in this study.

The latent transition analysis amongst substance use behaviors identified five behavioral profiles: (1) nonsmoker, infrequent binge drinker, (2) nonsmoker, infrequent binge drinker and cannabis consumer, (3) nonconsumer, (4) regular smoker and frequent cannabis consumer and (5) regular smoker. Nonconsumers, as well as regular smokers (with frequent or no cannabis consumption), did not change their profiles. Conversely, consumers involved in infrequent binge drinking changed their profiles during the COVID-19 pandemic. These findings support the notion of the clustering of substance use behaviors [6,39,40]. Previously, Lanza et al. [36] conducted an LTA on college students and reported four major profiles, including non-users, tobacco users, binge drinkers, and bingers with cannabis use, which are profiles similar to our findings. Risk behavior profiles over two semesters were found to be rather stable and showed that the highest percentage of college students were non-users and indicated the lowest change in the profile, including marijuana use [36]. In a previous study by Schweizer and colleagues [41], LTA was used to identify progression and reduction in smoking and alcohol use in University students. Findings of this study mainly suggested that, among young adults, both tobacco and alcohol use are temporally unstable behaviors, particularly among those using these substances at lower levels [41].

Sociodemographic characteristics (age, gender, relationship status), as well as interpersonal characteristics (depressive symptoms, being bored), were found to be associated with a change in engagement in HRB during the COVID-19 pandemic. With respect to age, older individuals were less likely to report increases in physical activity than younger individuals. Females were found to be more likely to increase their physical activity behaviors compared to males, highlighting potential differences in coping styles by gender during the COVID-19 pandemic.

A further factor associated with increased engagement in HRB during the COVID-19 pandemic was one’s relationship status. Those who reported being in a complicated relationship were more likely to report increased alcohol consumption and binge drinking, while for tobacco smoking, mixed findings were noted. Similarly, not having a trusted person (i.e., somebody to confide in and to discuss personal matters) was related to decreases in physical activity.

Reporting depressive symptoms was found to be associated with increased risk-taking behavior (number of cigarettes and drinks, binge drinking as well as cannabis use, physical activity), but equally associated with decreased risky drinking pattern (increases in physical activity) during the COVID-19 pandemic. It appears that some students with depressive symptoms reduced their engagement in HRBs, while others might have turned to HRB as a way of coping with potential additional stress perceived during the COVID-19 pandemic. Depressive symptoms were also found to be one of the most relevant factors associated with HRB among US-adults [38]. Moreover, a survey focusing on changes in alcohol use in US university students during the COVID-19 pandemic found that students with more symptoms of depression and anxiety reported greater increases in alcohol consumption [42]. Generally, those reporting depressive symptoms, those being bored or in unstable relationships, were disproportionally burdened by the COVID-19 pandemic with regard to their HRB.

### 4.1. Strengths and Limitations

To the best of the authors’ knowledge, this is the first study examining changes in HRB and their association with sociodemographic and interpersonal characteristics during the COVID-19 pandemic among the German student population. A range of HRB was investigated, allowing for the exploration of key risk behaviors of importance for the prevention of non-communicable diseases. LTA was used to identify profiles of HRB to account for clustering and correlation of single variables.

Some limitations of the study, however, need to be acknowledged. Foremost, the study was conducted with a convenience sample, which means that the results are not representative of the German student population as a whole. A gender imbalance was present in our sample, meaning that selection bias cannot be ruled out. Similarly, over a quarter of respondents were students of Medicine or health-related disciplines, which again limit the generalizability of the findings to the entire student population. The cross-sectional nature of the study with retrospective assessment of personal behaviors prior to COVID-19 may have been affected by recall bias. Moreover, certain covariates, such as depressive symptoms, were only assessed at one time point (during COVID-19), not allowing for a true pre-/post-pandemic situation comparison. Risk behavior was assessed using self-report measures, which, particularly with regard to physical activity behaviors, may lead to overestimation of one’s own behaviors. Similarly, the under- or over-reporting of other HRB cannot be ruled out. However, respondents were assured that the survey was anonymous, and web-based surveys have previously been found to be fairly reliable means of collecting HRB data among university students [43]. Binge drinking in this study was defined as six drinks or more for both female and male students; while in line with the cutoff point in the AUDIT guidelines, it is higher than the numbers used in other studies following a gender-specific approach and generally lower thresholds [33,44]. Finally, the survey was conducted in May 2020, at a time immediately following the first lockdown in Germany when some of the earlier restrictions had been lifted and announcements had been made about, for instance, the organization of the summer semester.

### 4.2. Future Research and Implications

In different populations, including university students, engagement in HRB has been shown to cluster [6,39]. The profile analysis indicated how smoking, binge drinking and cannabis use co-occurred. Profiles of the latent class analysis focused solely on substance use behaviors. Future research may extend this work and also investigate changes in additional risk behaviors, such as risky sexual and dieting behaviors, gambling or drug abuse.

As a risk factor for engagement in PA, gender differences could be observed. Gender differences have also been previously reported for engagement in HRB [34] and, in the case of the present study, might highlight potentially different ways of coping with the pandemic situation. Future research should focus on identifying coping strategies in order to identify subgroups of particular need for intervention in future pandemics. A range of other characteristics has been shown to be associated with engagement in HRB. Foremost, evidence suggests that one’s socioeconomic status is related to engagement in HRB [20]. For example, belonging to a high-income family has been reported as a risk factor for risky alcohol consumption [45]. It would be of interest to capture socioeconomic differences in engagement in HRB and changes in HRB during the COVID-19 pandemic. As socioeconomic status is related to risk behaviors, efforts that focus on structural and environmental factors have been argued to prove the most fruitful intervention strategies for behavior modification [21]. Additionally, characteristics associated with harmful alcohol intake have been living alone or in student accommodation [6] and a range of other individual and social factors [39] that could be investigated in future studies. Social factors, such as perceptions of peer drinking norms (e.g., drinking norms, social pressure), have previously been linked to engagement in HRB [22]. Whilst these were not investigated in the present study, it would be of interest to see whether, due to governmental restrictions regarding social gatherings, social pressures and perceptions of normative behavior changed among university students. Interestingly, a Belgian study conducted among the adult population reported that boredom was found to be the most often reported motive for an increase in tobacco and cannabis use during the COVID-19 pandemic, whereas conviviality was the main reason for alcohol consumption [46]. The present study focused on reported changes during the first wave of the SARS-CoV2 outbreak in Germany. It would be of value to assess engagement in HRB and changes in engagement with the continued COVID-19 pandemic and further waves and restrictions.

Universities and higher education have been previously emphasized as appropriate and promising settings for the implementation of lifestyle interventions [47,48]. In Germany, several programs were developed for the prevention and reduction of certain risk behaviors among university students that may also be used during the pandemic. Regarding substance use behaviors, stand-alone web-based electronic brief interventions [49] and electronic normative feedback programs [50] that incorporate data on substance use behavior at universities may be implemented. Physical activity may be promoted by developing online activities for courses normally offered on campus and by sending prompts to students through e-learning platforms [51].

In fact, the importance of establishing “health-promoting universities” in the sense of targeting health at the individual but also at the institutional level has been considered a promising way forward. Further initiatives to undertake health-promoting activities and to establish a health-promoting campus environment are warranted, given high levels of engagement in HRB among German university students. As such, university students as a whole present a defined and easy to identify the group at risk. As was shown in this study and other previous studies [1,52], health behaviors tend to cluster among university students, and, thus, initiatives to address multiple health risk behaviors rather than solely focusing on single behaviors may be valuable.

## 5. Conclusions

The current study provides the first results on the impact of the COVID-19 pandemic on German university students’ HRB and wellbeing. Engagement in HRB among university students remains common, with a wide majority of students reporting alcohol consumption and physical inactivity. While only a few changes were reported with regard to smoking and cannabis use, changes occurred with regard to binge drinking and, particularly, regarding PA. Sociodemographic as well as psychological characteristics, such as depressive symptoms, were related to changes in engagement in HRB. To conclude, in times of the COVID-19 pandemic, it remains important to monitor student health and to offer preventative public health interventions, targeting single as well as multiple health risk behaviors and their determinants in university students.

## Figures and Tables

**Table 1 ijerph-18-01410-t001:** Baseline characteristics of the study sample (N = 5021).

		**% Total**
**Sociodemographic information**	**Gender**	
	Female	69.4
	Male	29.4
	Diverse	1.2
	**Age Categories (years)**	
	17–18	1.6
	19–20	15.9
	21–22	24.2
	23–24	21.4
	>25	36.9
	**Place of birth**	
	In Germany	87.9
	Outside Germany	12.1
	**Relationship status**	
	Single	42.8
	In a relationship	53.3
	It is complicated	3.9
**Study-related information**	**Higher Education Institution**	
	Charité-Universitätsmedizin Berlin	14.7
	University of Bremen	37.6
	Heinrich Heine University Duesseldorf	12.2
	University of Siegen	34.4
	Other	1.1
	**First-year student**	22.9
	**Enrolled program**	
	Bachelor	53.8
	Master	22.7
	Doctoral	4.7
	State examination (medicine, law)	18.1
	Other	0.8
	**Field of study (*n* = 5017)**	
	Medicine/health sciences	25.4
	Other	74.6
**Further covariates**	**Trusted other (*n* = 4953)**	
	Yes	90.4
	No	9.6
	**Being bored (*n* = 4933)**	
	None of the time	40.2
	Some of the time	36.6
	Most/all of the time	23.7

**Table 2 ijerph-18-01410-t002:** Health risk behaviors (HRB) among German university students (in %).

	Smoking (*n* = 4950)		Binge Drinking (*n* = 4965)		Cannabis Use (*n* = 4939)		Vigorous Physical Activity (*n* = 4983)		Moderate Physical Activity (*n* = 4983)	
	Before COVID-19	During COVID-19	Before COVID-19	During COVID-19	Before COVID-19	During COVID-19	Before COVID-19	During COVID-19	Before COVID-19	During COVID-19
(Almost) none	80.6	82.4	54.2	75.3	89.2	91.2	13.3	21.7	9.3	10.1
(Less than) once a week	6.2	4.5	43.8	20.7	7.7	5.1	35.8	33.4	34.7	31.9
More than once a week	13.2	13.1	2.0	3.9	3.1	3.6	50.9	44.9	56.0	58.0

**Table 3 ijerph-18-01410-t003:** Change in HRB before and during the COVID-19 pandemic (in %).

	Number of Cigarettes	Number of Drinks	Binge Drinking	Cannabis Use	Vigorous Physical Activity	Moderate Physical Activity
Increase	4.4	18.4	5.4	2.8	19.3	23.3
No change	92.3	62.2	70.2	93.3	50.1	54.9
Decrease	3.3	19.4	24.4	3.9	30.6	21.8

**Table 4 ijerph-18-01410-t004:** Associations between change in risk behavior and age, gender, place of birth, relationship status, being a first-year student, study field, trusted other, boredom as well as depressive symptoms—Results of multinomial logistic regressions (ORs and 95% CI).

		Number of Cigarettes				Number of Drinks				Binge Drinking				Cannabis Use			
Variables		Increase vs. No Change		Decrease vs. No Change		Increase vs. No Change		Decrease vs. No Change		Increase vs. No Change		Decrease vs. No Change		Increase vs. No Change		Decrease vs. No Change	
		OR	95% CI	OR	95% CI	OR	95% CI	OR	95% CI	OR	95% CI	OR	95% CI	OR	95% CI	OR	95% CI
Age		1.05	1.02–1.07	1.02	0.99–1.05	1.00	0.98–1.01	**0.95**	**0.93–0.97**	1.01	0.99–1.04	**0.92**	**0.90–0.93**	0.99	0.96–1.03	**0.93**	**0.89–0.97**
Gender	Male	1.0		1.0		1.0		1.0		1.0		1.0		1.0		1.0	
	Female	0.94	0.68–1.29	0.82	0.58–1.16	**1.20**	**1.00–1.43**	0.87	0.74–1.03	1.00	0.75–1.35	**0.83**	**0.72–0.97**	0.95	0.64–1.40	**0.72**	**0.52–0.99**
	Diverse	1.29	0.44–3.76	1.99	0.68–5.81	1.36	0.71–2.60	0.63	0.28–1.46	1.41	0.56–3.52	0.59	0.27–1.29	0.55	0.72–4.10	2.08	0.79–5.52
Place of birth	Germany	1.0		1.0		1.0		1.0		1.0		1.0		1.0		1.0	
	Other	1.04	0.69–1.55	0.81	0.49–1.34	0.81	0.64–1.03	0.85	0.67–1.08	0.74	0.49–1.11	**0.65**	**0.52–0.82**	0.92	0.53–1.58	0.99	0.62–1.58
Relationship status	In relationship	1.0		1.0		1.0		1.0		1.0		1.0		1.0		1.0	
	Single	1.13	0.83–1.53	0.91	0.65–1.29	**1.23**	**1.04–1.44**	0.96	0.82–1.12	1.17	0.89–1.54	1.02	0.89–1.18	1.11	0.77–1.62	1.09	0.80–1.49
	Complicated	**2.03**	**1.15–3.57**	**2.35**	**1.29–4.29**	**1.56**	**1.07–2.28**	1.43	0.98–2.06	**1.77**	**1.03–3.04**	0.97	0.67–1.40	1.75	0.85–3.62	1.33	0.66–2.71
First-year student	No	1.0		1.0		1.0		1.0		1.0		1.0		1.0		1.0	
	Yes	0.78	0.55–1.13	1.10	0.75–1.60	0.94	0.78–1.13	0.99	0.83–1.18	1.00	0.74–1.37	0.89	0.75–1.04	0.64	0.40–1.02	0.89	0.62–1.26
Health-related study field	No	1.0		1.0		1.0		1.0		1.0		1.0		1.0		1.0	
	Yes	1.09	0.78–1.52	0.93	0.63–1.38	0.96	0.80–1.15	0.99	0.83–1.17	0.80	0.58–1.12	1.14	0.97–1.33	0.88	0.57–1.34	0.82	0.57–1.18
Trusted other	Yes	1.0		1.0		1.0		1.0		1.0		1.0		1.0		1.0	
	No	0.76	0.47–1.21	1.03	0.62–1.73	0.87	0.66–1.14	0.99	0.77–1.29	1.00	0.67–1.49	0.87	0.68–1.11	1.09	0.63–1.90	1.16	0.72–1.87
Time being bored	None	1.0		1.0		1.0		1.0		1.0		1.0		1.0		1.0	
	Some	0.97	0.69–1.36	1.02	0.69–1.52	1.00	0.84–1.19	0.94	0.79–1.11	**1.38**	**1.02–1.87**	0.96	0.82–1.12	1.19	0.79–1.78	1.19	0.85–1.67
	Most/all	1.16	0.81–1.68	**1.74**	**1.15–2.64**	0.89	0.72–1.09	0.94	0.77–1.15	1.20	0.85–1.70	0.95	0.79–1.14	1.09	0.68–1.74	0.95	0.63–1.43
Depressive symptoms		**1.10**	**1.07–1.14**	1.03	0.99–1.07	**1.07**	**1.05–1.09**	1.01	0.99–1.03	**1.11**	**1.07–1.14**	1.01	0.99–1.03	**1.07**	**1.03–1.11**	1.02	0.98–1.05

Numbers presented in bold reflect findings of statistical significance.

**Table 5 ijerph-18-01410-t005:** Associations between change in physical activity and age, gender, place of birth, relationship status, being a first-year student, study field, trusted other, boredom, as well as depressive symptoms—Results of multinomial logistic regressions (ORs and 95% CI).

		Vigorous Physical Activity				Moderate Physical Activity			
Variables		Increase vs. No Change		Decrease vs. No Change		Increase vs. No Change		Decrease vs. No Change	
		OR	95% CI	OR	95% CI	OR	95% CI	OR	95% CI
Age		**0.97**	**0.96–0.99**	1.00	0.99–1.02	**0.96**	**0.94–0.97**	1.01	0.99–1.02
Gender	Male	1.0		1.0		1.0		1.0	
	Female	**1.44**	**1.20–1.73**	0.91	0.78–1.05	**1.32**	**1.11–1.55**	**0.77**	**0.66–0.91**
	Diverse	0.80	0.34–1.88	0.88	0.48–1.60	0.76	0.36–1.63	0.57	0.29–1.13
Place of birth	Germany	1.0		1.0		1.0		1.0	
	Other	1.15	0.91–1.45	1.02	0.84–1.25	0.98	0.77–1.23	**1.53**	**1.24–1.88**
Relationship status	In Relationship	1.0		1.0		1.0		1.0	
	Single	0.92	0.78–1.08	**1.15**	**1.00–1.32**	1.14	0.98–1.33	0.98	0.84–1.15
	Complicated	0.77	0.51–1.17	0.96	0.68–1.35	0.70	0.47–1.06	0.82	0.57–1.19
First-year student	No	1.0		1.0		1.0		1.0	
	Yes	1.08	0.91–1.29	0.92	0.78–1.08	1.13	0.96–1.34	1.08	0.90–1.29
Health-related study field	No	1.0		1.0		1.0		1.0	
	Yes	1.07	0.90–1.27	1.08	0.93–1.27	**0.82**	**0.70–0.97**	1.17	0.98–1.39
Trusted other	Yes	1.0		1.0		1.0		1.0	
	No	0.97	0.74–1.29	1.06	0.85–1.33	1.06	0.81–1.38	**1.39**	**1.09–1.76**
Time being bored	None	1.0		1.0		1.0		1.0	
	Some	1.13	0.95–1.34	1.15	0.99–1.34	1.05	0.90–1.24	0.97	0.81–1.15
	Most/all	0.96	0.77–1.19	**1.42**	**1.19–1.70**	**1.23**	**1.01–1.49**	**1.35**	**1.11–1.64**
Depressive symptoms		**1.02**	**1.01–1.04**	**1.08**	**1.06–1.09**	**1.03**	**1.01–1.05**	**1.10**	**1.09–1.12**

Numbers presented in bold reflect findings of statistical significance.

**Table 6 ijerph-18-01410-t006:** Latent profile analysis of three HRBs (Smoking, binge drinking and cannabis consumption). Model selection and fit statistics (Bayesian information criterion), item-response probabilities (ρ-estimate), latent class characterization and matrix of transition probabilities.

Latent Profiles of Risk Behaviors					1	2	3	4	5	6
Model selection		BIC	DF	Frequency (%)						
Number of profiles	3	2630.23	702	Pre-COVID	0.348	0.160	0.491			
				During COVID	0.155	0.160	0.685			
	4	2452.35	689	Pre-COVID	0.352	0.102	0.474	0.071		
				During COVID	0.160	0.067	0.666	0.107		
	5	1768.61	674	Pre-COVID	0.341	0.047	0.477	0.035	0.100	
				During COVID	0.159	0.035	0.668	0.038	0.101	
	6	1794.25	657	Pre-COVID	0.331	0.041	0.482	0.033	0.099	0.015
				During COVID	0.159	0.018	0.668	0.032	0.099	0.024
**Item Response Probabilities**										
**Variable**		**Categories**		ρ-**Estimate**	**1**	**2**	**3**	**4**	**5**	
Smoking (avg. cigarettes/day)			none		0.970	0.822	0.992	0.294	0.015	
Binge drinking (frequency)			(almost) none		0.016	0.350	0.988	0.497	0.455	
Cannabis consumption (frequency)			(almost) none		0.965	0.000	0.995	0.000	0.909	
Smoking (avg. cigarettes/day)			1–9/day		0.027	0.178	0.007	0.599	0.719	
Binge drinking (frequency)			low		0.924	0.597	0.012	0.418	0.451	
Cannabis consumption (frequency)			low		0.034	0.992	0.004	0.144	0.072	
Smoking (avg. cigarettes/day)			10+/day		0.003	0.000	0.001	0.107	0.266	
Binge drinking (frequency)			high		0.060	0.053	0.000	0.085	0.093	
Cannabis consumption (freqquency)			high		0.001	0.008	0.000	0.856	0.019	
**Classes**	**Description**									
1 Class	Nonsmoker, infrequent binge drinker									
2 Class	Nonsmoker, infrequent binge drinker and cannabis consumer									
3 Class	Nonconsumer									
4 Class	Regular smoker and frequent cannabis consumer									
5 Class	Regular smoker									
**Transition Matrix**	**Frequencies**				**During COVID-19**					
					1 Class	2 Class	3 Class	4 Class	5 Class	
			**Pre COVID-19**	1 Class	0.434	0.015	0.543	0.001	0.006	
				2 Class	0.000	0.629	0.261	0.110	0.000	
				3 Class	0.019	0.000	0.976	0.000	0.004	
				4 Class	0.044	0.000	0.023	0.920	0.013	
				5 Class	0.000	0.000	0.036	0.000	0.964	

BIC: Bayesian information criterion, DF = degrees of freedom.

## Data Availability

Due to the nature of this research, participants of this study did not agree for their data to be shared publicly, so supporting data are not publicly available. Data are available on request from the corresponding author for collaborating researchers within the C19 ISWS consortium, as consent for this was provided from all participants.

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
