# Peer review of "Engagement in Health Risk Behaviours before and during the COVID-19 Pandemic in German University Students: Results of a Cross-Sectional Study"

_ijerph, 2021, doi:10.3390/ijerph18041410_

Round 1
Reviewer 1 Report
A well written paper that should be improved in the conclusions section with a separation of the implications for future research and health promotion practice. Beyond a mention of ¨health promoting universities¨ the authors should identify relevant interventions or programs that would be effective, given the evidence from the study to reduce HRB, for German University Students. This is an essential revision before publication.
Reviewer 2 Report
This is a report on a survey of health risk behaviours that was conducted online with a convenience sample of students, obtaining retrospective self-reports of behaviour and change. There are serious limitations on the quality of such data, but these are fairly presented in the ‘Strengths and limitations’ section.
My main criticism concerns the presentation of the LTA. (Incidentally, note that the LTA method has not been referenced in the paper; the reference [34] concerns LPA only.) LTA directly models changes in behaviour, which is the main interest of the paper, so why has it been relegated to such a secondary position in the paper (one short paragraph and a supplementary table)? I believe that these results should be brought into the main paper. In particular, they show quite clearly a main (although unsurprising) finding, namely, that the only common transitions are from groups 1 and 2 to 3.
Concerning the conduct of the survey, I would like the authors to explain what information was given to students in the invitation to participate. This could very well have an influence on who chose to take part, and hence on possible biases in the sample. Regarding the achieved sample, what is the explanation of the very large gender imbalance? Furthermore, the sample size per university should be given, as an absolute number and as a percentage of eligible students. (We are told that the target of 10% was achieved in three of the four universities, but if the percentage was, say, 25% in one and 5% in another, I would want to know why there was such a big difference.)
The paper is quite clearly written; suggested minor edits are listed below.
Page 2 / line 2: “have been brought in connection” – write “into” instead of “in”
2 / 12: “More than 70%….”. I found this sentence unclear. I think it means that “More than 70% of students remained non-users of cannabis throughout their studies or kept cannabis use at a low level (10%)”
2 / 23: “is prevalent among students” – “are” instead of “is”
2 / 5 (of Section 1.2); “Examples of”
2 / 7 from end: “were previously been found” – delete “been” or change “were” to “have”
2 / next to last: “particular” not “particularly”?
2 / Title of Section 1.3: a title should not appear as the last line on a page
3 / 1-2: “pandemic have had and continue to have”
3 / 2 of second paragraph: “impact of the 2008”
3 / 1 of third para: “on university students”
3 / 3 of third para: “adults in the general population”
4 / end of 1st para of Section 2.3: “had to change their offers” – unclear. Does it mean “had to change their way of working”?
4 / Section 2.4: give reference for Qualtrics
5 / 1 of Section 2.6: “interest for HRB were”
5 / 2 of para concerning number of drinks per week: “prior” should be “prior to” or “before”
5 / end of page: subtitle “Covariates” should not appear as the last line on a page
6 / next to last: “completed the survey”
Table 2: Data on Number of drinks have been omitted
8 / Title of Section 3.3: correct the spelling of “engagement”
8 / 2 of Section 3.3: “equally” is probably intended to mean “also” but it is an unfortunate choice of word because in fact the effects in the ‘decrease’ direction seem to be smaller than in the ‘increase’ direction. I suggest “also”.
8 / 7 from end: delete comma after “both”
9 / 2: “were not found”
10 / 12 from end: ”in line with previous studies”
12 / 1-4: I found this sentence unclear. I suggest changing “where at least some prior restrictions were lifted” to “when some of the earlier restrictions had been lifted”, and “first information…..were announced” to “announcements had been made about, for instance, the organization of the summer semester”. However, I didn’t really understand the importance of the last part of the sentence.
12 / 9 of para beginning “Particularly”: “as a risk factor”
